# Changes in the Ecological Status of Rivers Caused by the Functioning of Natural Barriers

Katarzyna Połeć [1], Antoni Grzywna [1,*], Monika Tarkowska-Kukuryk [2] and Urszula Bronowicka-Mielniczuk [3]

1 Department of Environmental Engineering and Geodesy, University of Life Sciences in Lublin, Leszczyńskiego 7, 20-069 Lublin, Poland; katarzyna.polec@up.lublin.pl
2 Department of Hydrobiology and Protection of Ecosystems, University of Life Sciences in Lublin, B. Dobrzańskiego 37, 20-262 Lublin, Poland; monika.kukuryk@up.lublin.pl
3 Department of Applied Mathematics and Computer Science, University of Life Sciences in Lublin, Głęboka 28, 20-612 Lublin, Poland; urszula.bronowicka@up.lublin.pl
* Correspondence: antoni.grzywna@up.lublin.pl

**Abstract:** Introducing the European beaver to the catchment area, which adjusts the habitat to its own needs (by building dams), may have a positive impact on the ecology, geology, and hydromorphology of rivers and intensify the water self-purification process. In this study, a comparative assessment of the ecological status was made between the areas where the species *Castor fiber* L. occurs (habitat type A) and the areas unaffected by the influence (habitat type B). For this purpose, the Macrophyte River Index (MIR) and the Hydromorphological River Index (HIR) were calculated, along with the floristic indicators of biodiversity: species richness and Margalef, Shannon–Wiener, and Simpson indices. Only 35% of the sites met the standard of good ecological status. The presence of hypertrophic species and anthropogenic modifications of the river bed had a negative impact. The spread of beavers has a significant positive effect on changes in hydromorphological conditions and water levels in the river. The water levels in habitat types A and B were 0.504 and 0.253 m, respectively. There were statistically significant differences in the HIR values between habitat types A and B, which were 0.585 and 0.535, respectively. In habitats of type A, the heterogeneity of the current and bed material as well as the diversity of elements accompanying the tree stands increased. Research has shown greater species richness and greater biodiversity of macrophytes in the habitats of beaver dams. The research confirmed the significant influence of the European beaver on changes in the environment. The activity of beavers intensifies the processes of introducing wetland and rush species to forest areas.

**Keywords:** beavers; macrophytes; biodiversity; hydromorphology; protected area





## 1. Introduction

Rivers are among the most transformed ecosystems. Changes in the morphology of the river beds were made as early as in the tenth century by building systems of small hydrotechnical structures—mills and weirs. The later deforestation of floodplains and their agricultural use and riverbed regulation caused irreversible changes in the water relations of the catchment area [1]. Continuous demographic growth and urban development are causing further changes in land use. The urbanization of the catchment area contributes to the degradation of watercourses and water reservoirs [2]. Additionally, climate changes affecting the amount of rainfall and causing a more frequent occurrence of extreme phenomena, such as floods and droughts, create the need to search for new water retention methods [3]. For many years, actions have been taken to naturalize the catchment area, restore its original ecological state, increase retention, and improve water quality. These activities include, inter alia, the introduction to the catchment area of a species whose habitat activities have positive benefits for the ecology of river corridors [4].

Beavers are called ecosystem engineers and build dams on small rivers in wetland habitats [5]. Medium-sized dams built by beavers are impressive, but even they are smaller

than the world's largest dam, 850 m long, discovered in Canada. The Alberta Dam is located in Canada's Wood Buffalo National Park and was discovered by ecologist Jean Thie looking for signs of climate change using satellite scanning [6]. In Poland, the restoration of a healthy population of the European beaver (*Castor fiber* L.) had a purely biological basis, as it was concluded that a small number of beavers might be insufficient to ensure the survival of this species [7]. Due to many years of the reintroduction and protection of beavers throughout the country, it is currently estimated that the number of individuals of this species in Poland exceeds 127,000 individuals [8]. Beavers transform the natural environment, having a huge impact on the ecology, geology, and hydromorphology of the habitats in which they occur. By their activity, beavers contribute to the generation of damage to the stand, agriculture, and water management. In the spring and summer months, beavers graze the tree stand, intensively dig burrows, and the dams they build contribute to the flooding of agricultural areas [9]. The negative activity of beavers creates many conflicts on the beaver–human line. The species is perceived mainly through the prism of its negative impact on the environment, but its presence is widely tolerated by European society [10]. The presence of natural barriers on rivers causing the formation of beaver ponds contributes to the dispersion of pollutants resulting from the agricultural use of the catchment area [11]. The nutrient content in surface waters changes spatiotemporally [12]. The concentration of nutrients, pH, and the velocity of water shape the development of aquatic vegetation [13]. The slowing down of the water flow by beavers influences the diversity and complexity of macrophyte communities on the riverbed [11]. Therefore, the Macrophyte River Index (MIR) can be used to determine the impact of beaver dams on changes in the ecological status of rivers. It is a biological method of assessing the ecological status of river waters based on the requirements of the Water Framework Directive [14]. Other systems of macrophyte indicator species are also used to assess the ecological status of rivers. In Poland, the most frequently used are IBMR, MTR, RMNI, ITEM, and RI [15–17].

The ecological condition of rivers is not only determined by biological elements. River regulation to address flood problems has a negative impact on hydromorphology. Human intervention in the course and simplification of the river pattern often leads to the loss of geomorphological diversity; the biodiversity within the river bed also decreases, and the hydromorphological continuity is disturbed [18,19]. There is a relationship between the biological components of the aquatic ecosystem and the hydromorphological conditions in rivers, as the increase in hydromorphological diversity affects the species richness of aquatic life [20]. The use of the Hydromorphological River Index (HIR) method allows for the assessment of surface water bodies in terms of considering the need for their reclamation [21].

This study aimed to assess the hydromorphological state in terms of the diversification of land use and the conditions of the protected species significantly affecting the transformation of the river bed morphology and water conditions. Moreover, the aim of the study was also to assess the ecological status of surface waters on the basis of the Macrophyte River Index. A comparative assessment of the hydromorphological and ecological status was carried out between the areas of occurrence of *Castor fiber* L. (10 sites) and the areas not affected by the European beaver (10 sites). In recent years, many projects have been carried out to improve the ecological conditions and restore the natural character of rivers and wetlands. So far, there has been little information on the success of the implemented measures. As the research on the impact of beavers' activity has not been conducted on a large scale so far, the focus should be on the analysis of the species' impact on the environment in terms of enhancing the ecological status. Determining the impact of the activity of a protected species on the ecological status and biodiversity may be of significant importance in modeling programs for the restoration of catchment areas and creating plans to increase natural water retention and protection.

## 2. Materials and Methods

### 2.1. Study Area

The research was carried out in areas with similar land use. The research area covers protected areas (Polesie National Park, Roztocze National Park). The Polesie National Park (PPN) is located in the Western Polesie macroregion and was created mainly to protect wetlands. The characteristic of the Polesie National Park is the presence of three complexes of open peat bogs and a number of small mid-forest bogs. Open peatland constitutes 16.5% of the park's area. Forest land is the predominant type of ecosystem and the predominant element of the ecological landscape. Deciduous species dominate within this group. Roztocze National Park (RPN) was established to protect diverse forest ecosystems, covering 93.81% of the park's area. *Pinus sylvestris* L. (35.43%), *Fagus sylvatica* L. (22.02%), and *Abies alba* Mill. (16.13%) dominate among the species in the forest ecosystems. The selected research points were located on the following watercourses: Włodawka, Mietiułka, Piwonia, Tyśmienica, Świerszcz and Szum. The studied rivers were similar in terms of depth, velocity, track width, type of bottom substrate, and water quality. In all sites (catchment area 40–100 km², river path width 3–7 m, water level 0.3–0.7 m), the dominant bottom substrate was sand with silt, laminar flow. The activity of the European beaver was the factor differentiating the sites. The research sites were selected in such a way that 10 of them were related to the location of beaver dams (habitat type A). The remaining 10 sites were located in the areas where the presence of European beavers was not found, at a distance of 500 m below the beaver dam (habitat type B).

Samples of aquatic macrophytes were collected in small rivers located in protected areas. The geographic location of the research sites was determined using GPS, and their coordinates, distribution, and range of influence using Google Earth. Finally, the analysis of the results was performed for 20 sites, which were located at a distance of at least 5 km from each other. Aquatic plants were collected for research in May and September 2021, with 12 sites located in the Polesie National Park and 8 sites located in the Roztocze National Park.

### 2.2. Macrophytes Survey

At each site, macrophytes were examined on a 100 m section of the watercourse; 10 transects were determined every 10 m, which allowed for a total of 10 samples from each site. The coverage of each species was determined using the 10-point Braun-Blanquet scale [22]. Following this, the Macrophyte River Index (MIR) was calculated.

The value of the MIR reflects the ecological state of the river depending on the degree of trophic degradation of the river and tolerance to environmental conditions [23]. The value of the MIR indicator was calculated on the basis of the following formulas:

$$\text{MIR} = \frac{\sum(L_i \times W_i \times P_i)}{\sum(W_i \times P_i)} \times 10 \tag{1}$$

where $L_i$—trophy number for the $i$ species, $W_i$—weighting factor for the $i$ species, and $P_i$—coverage factor for the $i$ species.

$$MIR_{WJE} = \frac{MIR_{obl.}}{MIR_{ref.}} - 0.1 \tag{2}$$

where $MIR_{WJE}$—Macrophyte River Index expressed as a Factor Quality Ecological, $MIR_{obl.}$—MIR calculated for a given position, and $MIR_{ref.}$—reference MIR for a given river type.

On the basis of the limit values of the MIR coefficient for research sites (small lowland rivers), one of five ecological status classes was assigned [24].

Species identification of macrophytes was made on the basis of Bernatowicz and Wolny [25] and Szoszkiewicz et al. [26]. On the basis of functional groups, we recorded all species among four groups of macrophytes (emergent, submerged, floating, and eloides) [27].

On the basis of the community of macrophytes, the diversity indexes [28–31] were calculated. The coverage of each species was determined using the 10-point Braun-Blanquet scale [22]. The following transformation was applied: 1—0.1%, 2—0.5%, 3—1.75%, 4—3.5%, 5—7.5%, 6—17.5%, 7—32.5%, 8—50%, 9—67.5%, 10—87.5%.

Species richness (S) is the number of species in a community, habitat, or site. The Shannon–Wiener index (H) is expressed by the formula:

$$H = - \sum_{i=1}^{S} p_i ln p_i \tag{3}$$

$$p_i = \frac{n_i}{N_i} \tag{4}$$

where $n_i$—the number of individuals of a specific species, $N_i$—the number of all individuals of all species, and $p_i$—share of the $i$th species. The index reaches its highest values when the share of species is even, i.e., when all species have the same $p_i$.

The Simpson index (SDI) is calculated from the formula:

$$SDI = 1 - \frac{\sum n(n-1)}{N(N-1)} \tag{5}$$

Margalef Index (MI):

$$MI = \frac{S-1}{ln N} \tag{6}$$

where: $S$—number of all species, and $N$—the abundance of individuals expressed as a percentage.

### 2.3. Hydromorphological Research

Hydromorphological studies of rivers were carried out on the basis of the British River Habitat Survey (RHS) method [32]. It is a system for assessing the nature of the habitat and the quality of watercourses using morphological parameters and selected hydrological parameters. The assessment of the condition of the rivers was made on the basis of the Hydromorphological River Index, which allows for the valorization and classification of flowing waters [33]. The analyzed parameters included the physical attributes of the riverbed, types of vegetation in the riverbed, land use, vegetation structure, hydromorphological units, bank cross-sections, the occurrence of hydrotechnical structures, assessment of tree cover and shade, the width of the unused coastal zone, types of use of the valley, and occurring anthropogenic pressures. Based on the hydromorphological data, two indicators were calculated: Habitat Quality Assessment (HQA)—habitat naturalness index, and Habitat Modification Score (HMS)—habitat transformation index. The HIR multimetric was calculated with the formula:

$$HIR = \frac{\left(\frac{HQA-HMS}{100}\right) + 0.85}{1.8} \tag{7}$$

High HQA values indicate the great diversity of the landscape in the vicinity of the river. High HMS values indicate bank resection and construction work related to river engineering. For each research point, one in five classes of hydromorphological status was assigned in accordance with the limit values of the HIR multimeter for lowland rivers with a bed width of ≤30 m [24].

Measurements of water levels (WL) in the river were also made.

### 2.4. Statistical Analysis

In order to compare the distribution of the analyzed parameters with the division into habitat types A and B, boxplot plots were made.

Non-metric multidimensional scaling (NMDS) was conducted to examine the relationship between species composition with HIR and WL, assessing Bray–Curtis dissimilarity. We performed the NMDS using the metaMDS function of the 'vegan' package [34] in the R environment.

The Pearson correlation coefficient was calculated for the analyzed factors, and the significance test of correlation was used to evaluate the obtained results. The parametric t-test for dependent samples was used to evaluate the mean values of the analyzed parameters for the two site types. The results of the research were statistically analyzed using the Statistica 13 software.

## 3. Results and Discussion

On the basis of the field studies, the rivers were found to have a total of 43 macrophyte species. Among the total abundance of aquatic macrophytes, in the habitats of type A there were 36 species, and in the habitats of type B there were 32 species. Despite different physiographic conditions, 23 plant species were found in both types of habitats (Table 1). The abundance of aquatic macrophytes species for habitat types A and B was, on average, 6.7 and 5.5, respectively (Tables 2 and 3). For type A habitats, the species abundance ranged from 4 to 12 species, while for type B habitats from 3 to 8 species (Figure 1). The abundance of macrophyte species did not differ significantly between A and B habitats (df = 9, t = 2.092, $p = 0.066$) (Table 4). The modification of the river paths, whose shape resembles trapezoidal ditches, has a negative impact on the number of species. Other factors influencing the communities of aquatic macrophytes were the velocity and volume of the flow as well as bottom siltation [35]. In artificial watercourses in protected areas, species abundance ranged from 2 to 12 (5.5 on average) [36]. Despite research conducted in protected areas, the diversity of aquatic macrophyte communities was small. In England, Slovenia, and Slovakia, researchers found comparable species richness [37–39]. The most common species are the pleustophytes *Lemna minor* (12 sites) and *Lemna trisulca* (8), as well as the helophytes *Scirpus sylvaticus* (12) and *Phragmites australis* (9) (Table 1). Moreover, these species were characterized by high coverage, exceeding 60% of the site area. The common occurrence of *L. minor* and *L. trisulca* is associated with stagnant water or laminar flow in the watercourse [40]. *Lemna* sometimes forms large, compact lobes at a limited flow velocity and restricts the inflow of light and oxygen to deeper layers of water [27]. *L. minor* is often used to remove organic pollutants in municipal wastewater treatment plants in rural areas [41]. The common *P. australis* [34] is also used to remove nutrients and heavy metals from wastewater. Both *L. minor* and *P. australis* can be used as renewable energy sources [42,43]. The greatest expansion of *P. australis* occurs in drained wetlands due to the increase in salinity [44]. The invasion of this species occurs fastest when the water levels in the river are low. In our research, the height of plants very often exceeds 1.5 m, and the community sometimes covers the entire length of the riverbank at a given site. High and dense patches of *P. australis* shade the river path, which sometimes creates monocultures. Another frequently occurring species was *S. sylvaticus*, which occurs naturally in the wetlands of northern Europe [45–47]. This species can also be used for the accumulation of nutrients in wastewater treatment plants [48]. As in the case of *P. australis*, the occurrence of *S. sylvaticus* is associated with the low water level in the river. Species such as *L. minor*, *P. australis*, and *S. sylvaticus* have a wide range of ecological tolerance and cannot be used as bioindicators. The studies showed the presence of only one species with a narrow range of tolerance (*Stratiotes aloides*) and 11 species with a medium range.

**Table 1.** Occurrence of plant species at habitats—number of sites (abbreviations used in figures).

| Taxa | Habitat A | Habitat B | Total |
|---|---|---|---|
| *Cladophora* sp. (*Cla* sp.) | 1 | 0 | 1 |
| *Oedogonium* sp. (*Oed* sp.) | 1 | 0 | 1 |
| *Spirogyra* sp. (*Spi* sp.) | 1 | 1 | 2 |

**Table 1.** *Cont.*

| Taxa | Habitat A | Habitat B | Total |
|------|-----------|-----------|-------|
| *Ulothrix* sp. *(Ulo* sp.*)* | 1 | 0 | 1 |
| *Leptodictyum riparium (Lep_rip)* | 1 | 1 | 2 |
| *Platyhypnidium riparioides (Pla_rip)* | 1 | 1 | 2 |
| *Batrachium aquatile (Bat_aqu)* | 3 | 0 | 3 |
| *Berula erecta (Ber_ere)* | 1 | 2 | 3 |
| *Butomus umbellatus (But_umb)* | 1 | 0 | 1 |
| *Callitriche* sp. *(Call* sp.*)* | 0 | 1 | 1 |
| *Caltha palustris (Cal_ pal)* | 0 | 1 | 1 |
| *Carex gracilis (Car_gra)* | 0 | 1 | 1 |
| *Ceratophyllum demersum (Cer_dem)* | 0 | 1 | 1 |
| *Eleocharis palustris (Ele_pal)* | 0 | 1 | 1 |
| *Elodea canadensis (Elo_can)* | 1 | 1 | 2 |
| *Equisetum fluviatile (Equ_flu)* | 1 | 0 | 1 |
| *Equisetum palustre (Equ_ pal)* | 2 | 1 | 3 |
| *Galium palustre (Gal_pal)* | 2 | 1 | 3 |
| *Hydrocharis morsus-ranae (Hyd_mor)* | 2 | 1 | 3 |
| *Hydrocotyle vulgaris (Hyd_vul)* | 0 | 1 | 1 |
| *Iris pseudacorus (Iri_pse)* | 2 | 1 | 3 |
| *Lemna gibba (Lem_gib)* | 1 | 1 | 2 |
| *Lemna minor (Lem_min)* | 7 | 5 | 12 |
| *Lemna trisulca (Lem_tri)* | 4 | 4 | 8 |
| *Mentha aquatica (Men_aqu)* | 2 | 0 | 2 |
| *Myosotis palustris (Myo_pal)* | 2 | 0 | 2 |
| *Myosoton aquaticum (Myo_aqu)* | 0 | 1 | 1 |
| *Myriophyllum spicatum (Myr_spi)* | 1 | 0 | 1 |
| *Nuphar lutea (Nup_lut)* | 1 | 0 | 1 |
| *Phragmites australis (Phr_aus)* | 4 | 5 | 9 |
| *Potamogeton berchtoldii (Pot_ber)* | 1 | 1 | 2 |
| *Potamogeton crispus (Pot_cri)* | 1 | 0 | 1 |
| *Ranunculus repens (Ran_rep)* | 2 | 1 | 3 |
| *Rorippa amphibia (Ror_amp)* | 1 | 1 | 2 |
| *Rumex hydrolapathum (Rum_hyd)* | 4 | 2 | 6 |
| *Salix* sp. *(Sal_sp.)* | 1 | 0 | 1 |
| *Scirpus sylvaticus (Sci_syl)* | 7 | 5 | 12 |
| *Sium latifolium (Siu_lat)* | 1 | 3 | 4 |
| *Spirodela polyrhiza (Spi_pol)* | 3 | 3 | 6 |
| *Typha latifolia (Typ_lat)* | 0 | 1 | 1 |
| *Urtica dioica (Urt_dio)* | 1 | 5 | 6 |
| *Veronica beccabunga (Ver_bec)* | 1 | 1 | 2 |
| *Stratiotes aloides (Str_alo)* | 1 | 1 | 2 |
| **Species richness** | **36** | **32** | **43** |

**Table 2.** Basic descriptive statistics for dependent samples.

| Parameters | | Mean | Standard Deviation | Standard Error of the Mean |
|------------|------|------|--------------------|----------------------------|
| P 1 | HIR_A | 0.585 | 0.085 | 0.027 |
|     | HIR_B | 0.535 | 0.088 | 0.028 |
| P 2 | MIR_A | 0.688 | 0.120 | 0.038 |
|     | MIR_B | 0.670 | 0.171 | 0.054 |
| P 3 | H_A | 1.798 | 0.337 | 0.106 |
|     | H_B | 1.625 | 0.362 | 0.114 |
| P 4 | MI_A | 3.644 | 1.151 | 0.364 |
|     | MI_B | 3.067 | 1.002 | 0.316 |
| P 5 | SDI_A | 0.844 | 0.053 | 0.017 |
|     | SDI_B | 0.815 | 0.082 | 0.025 |
| P 6 | S_A | 6.70 | 2.263 | 0.715 |
|     | S_B | 5.50 | 1.841 | 0.582 |
| P 7 | LW_A | 0.504 | 0.200 | 0.063 |
|     | LW_B | 0.253 | 0.054 | 0.017 |

**Table 3.** Pearson's correlation coefficients.

| Dependent Attempts | N | Correlation | Significance |
|---|---|---|---|
| HIR_A and HIR_B | 10 | 0.983 | **0.000** |
| MIR_A and MIR_B | 10 | 0.637 | **0.048** |
| H_A and H_B | 10 | 0.688 | **0.028** |
| MI_A and MI_B | 10 | 0.532 | 0.114 |
| SDI_A and SDI_B | 10 | 0.667 | **0.035** |
| S_A and S_B | 10 | 0.627 | 0.052 |
| LW_A and LW_B | 10 | 0.120 | 0.742 |

Significant differences are shown in bold.

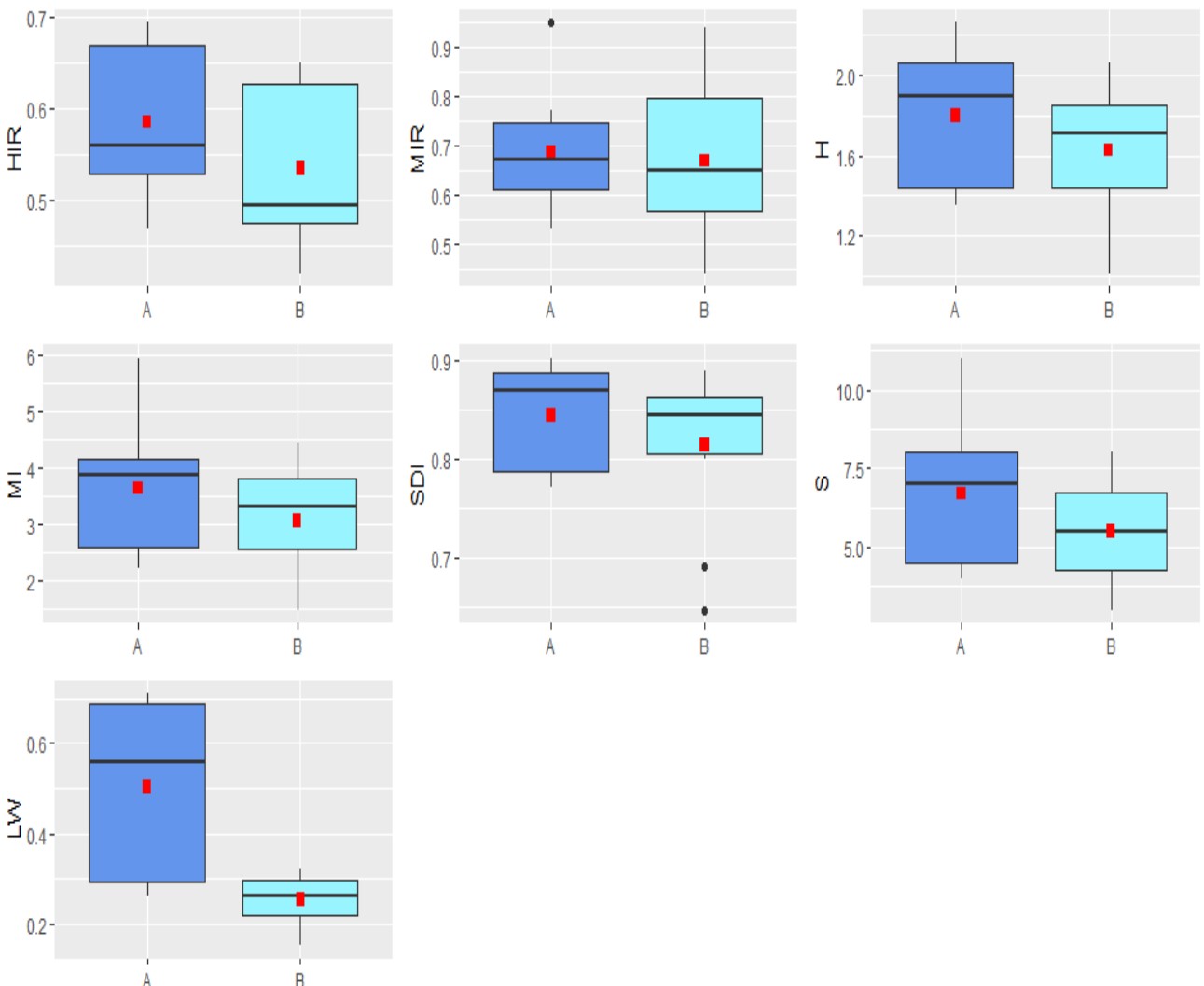

**Figure 1.** Box plots for the analyzed indicators. The box represents the first and third quartiles. The horizontal line across the central region of the box represents the median. A red square marks the mean value of the data. The whiskers are drawn to the most extreme observations located no more than 1.5 times the interquartile range away from the box. Any observation not included between the whiskers is considered an outlier and is plotted with a black dot. The whiskers indicate the minimum and maximum values when there are no outliers. A–habitat type A; B–habitat type B.

**Table 4.** Parametric *t*-test for dependent samples.

| Dependent Attempts | Differences in Dependent Samples | | | | | t | Significance |
|---|---|---|---|---|---|---|---|
| | Mean | Standard Deviation | Standard Error of the Mean | 95% Confidence Interval | | | |
| | | | | Lower Limit | Upper Limit | | |
| HIR_A and HIR_B | 0.050 | 0.016 | 0.0051 | 0.0391 | 0.0620 | 9.970 | **0.000** |
| MIR_A and MIR_B | 0.017 | 0.132 | 0.0418 | −0.0768 | 0.1126 | 0.428 | 0.679 |
| H_A and H_B | 0.173 | 0.277 | 0.0878 | −0.0256 | 0.3716 | 1.970 | 0.080 |
| MI_A and MI_B | 0.577 | 1.050 | 0.3322 | −0.1745 | 1.3285 | 1.737 | 0.116 |
| SDI_A and SDI_B | 0.029 | 0.061 | 0.0193 | −0.0137 | 0.0735 | 1.549 | 0.156 |
| S_A and S_B | 1.200 | 1.813 | 0.5735 | −0.0973 | 2.4973 | 2.092 | 0.066 |
| LW_A and LW_B | 0.251 | 0.201 | 0.0636 | 0.1071 | 0.3948 | 3.946 | **0.003** |

Significant differences are shown in bold.

The least frequent taxa are algae only found in one site, type A. The presence of algae is influenced by the inflow of light and the lack of water turbulence. Algae are able to assimilate nitrogen and phosphorus and reduce $CO_2$ emissions [49]. Some species of algae are also used in medicine [50]. High temperature and hypoxia can lead to harmful algae blooms, negatively affecting animals [51].

The occurrence of aquatic macrophytes is influenced by the water level in the river, the use of the surrounding area, and the hydromorpholophilic conditions (Figures 2–4). *Veronica beccabunga*, *Rorippa amphibia*, and *Carex gracilis* are associated with low water. *Potamogetoncrispus*, *Elodeacanadensis*, and *Spirodelapolyrhiza* occur mainly at high water levels. There was no occurrence of *E. canadensis* in watercourses with large fluctuations in water level and with a dynamic flow [52].

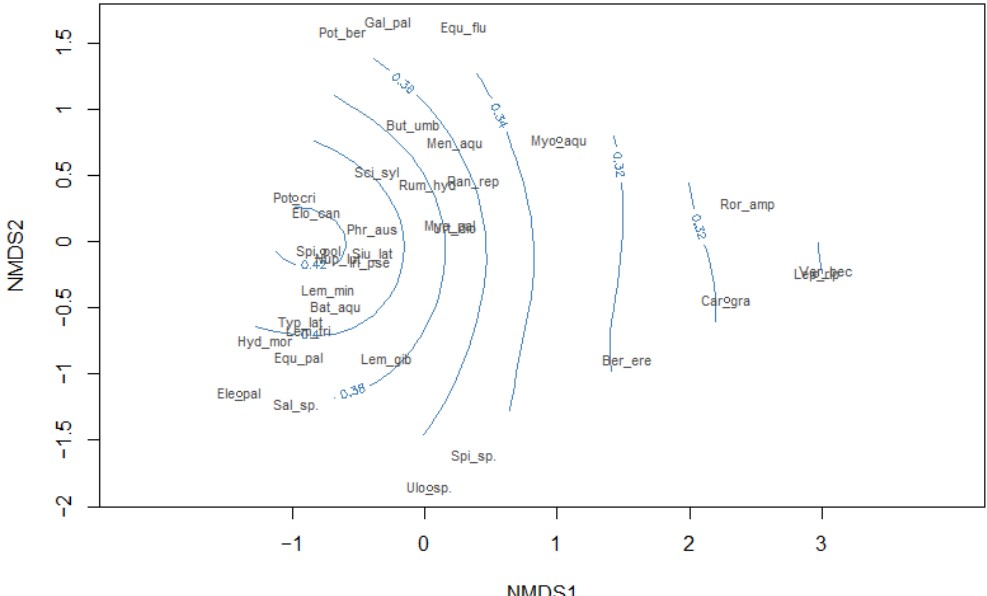

**Figure 2.** Influence of the water level on biodiversity. WL ranges at 0.32 to 0.44 m.

Most of the test sites are located in forest areas (75%), and only some sites of the B rivers are located in the vicinity of meadows. In these sites, the most common species are *E. canadensis*, *S. polyrhiza*, and *Hydrocharis morsus-ranae*. *E. canadensis* prefers sunlit locations with high water levels, and occurs most often in ponds and lakes and rarely in rivers.

The species composition of macrophytes depends on the hydromorphological conditions, mainly the width of the flow path, slope inclination, and bottom siltation. In rivers characterized by steep slopes and the deposition of mud on the bottom, *Batrachium aquatile*,

*Galium palustre*, and *Ranunculu srepens* are the most common. *Urtica dioica*, *Typha latifolia*, and *Calla palustris* play a dominant role in type B habitats (Table 1).

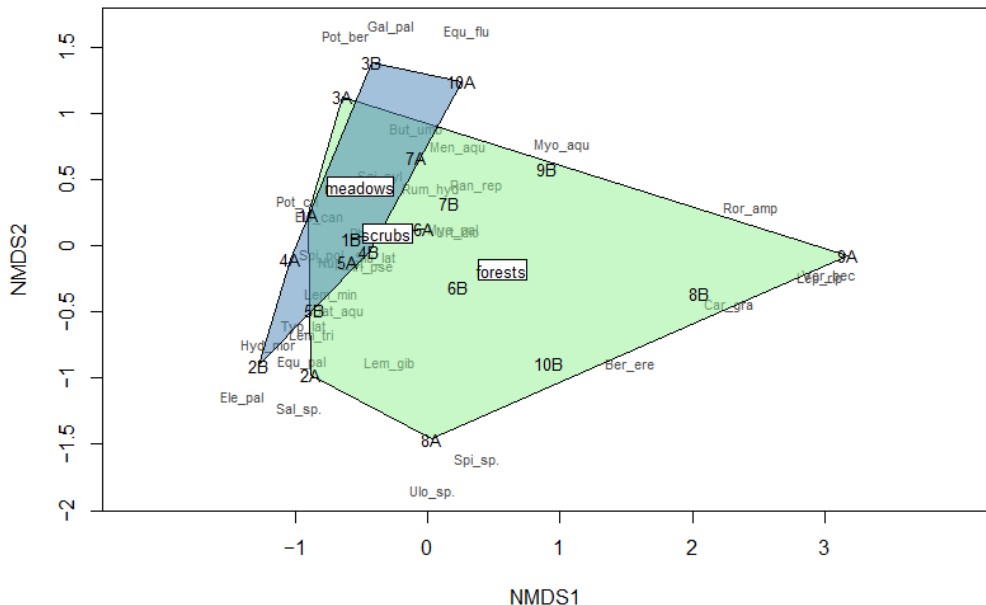

**Figure 3.** The impact of land use on the occurrence of plant species.

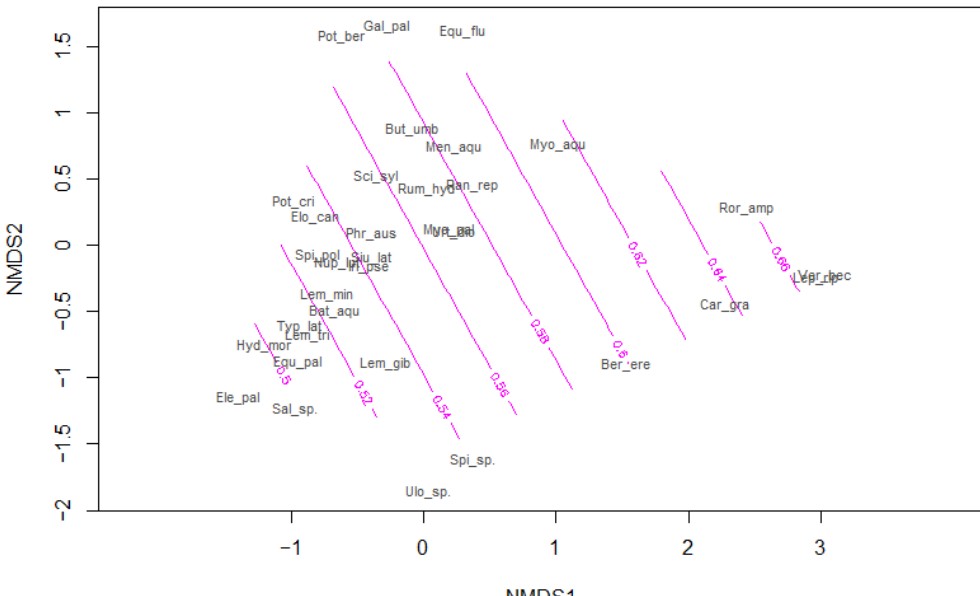

**Figure 4.** The influence of HIR on the occurrence of plant species. The HIR ranges from 0.50 to 0.66.

The Shannon–Wiener index indicates a range of 1 to 3 points as moderately polluted. Generally speaking, the species richness of macrophytes and the Shannon–Wiener diversity were higher in the sections with the beaver dam than in the neutral sections. However, the observed differences were not statistically significant (Table 4).

For a community dominated by one species, the value of the Simpson index as a dominance measure is 1. To avoid any misunderstandings in our analysis, we used it as a measure of equality by subtracting the dominance value from 1 [53]. Our research shows that the Simpson index as a measure of equality for A and B habitats was 0.845 and 0.815, respectively. The Margalef index for macrophytes identified in small watercourses is 3.64 and 3.07 for habitat types A and B, respectively.

The values of the analyzed biodiversity indicators (H, SDI, MI) are much higher than in wetlands [54]. The biodiversity of the Moselle floristic community was influenced by factors such as water level, shade, and the width and stability of the flow path [55].

The species diversity of aquatic macrophyte communities is also reflected in the ecological state of water expressed with the MIR [24]. MIR values for type A habitats average 0.688 and range from 0.55 to 0.77. Type B habitats are characterized by greater variability, where the MIR is 0.67 on average and ranges from 0.44 to 0.94 (Figure 1, Table 2). MIR values are statistically significantly correlated with each other (Table 3) and do not show significant differences between each other (Table 4). Based on the MIR value, 4 sites were classified as having very good ecological status (1st class), 5 sites as good (2nd class), and 11 sites as having moderate ecological status (3rd class). The ecological status was negatively affected by the presence of *L. minor* and *L. trisulca* as well as *Rumex hydrolapathum* and *S. polyrhiza*, most often in the same sites (Table 1). These species are characteristic of hypertrophic habitats. However, *L. minor* and *R. hydrolaphathum* have a wide range of ecological tolerance and may occur in other habitat types. The presence of *B. aquatile* and *Platyhypnidium riparioides* positively affected MIR values in type A habitats. These species are characteristic of mesotrophic habitats. The abundance of macrophytes depends on many factors: water flow, shading of the bed, nutrient concentration, and hydromorphological transformations. The differentiation of macrophytes in rivers is particularly influenced by the speed of water flow in the riverbed [56]. Previous studies present contradictory results regarding the impact of beaver invasions on the community of aquatic macrophytes [57]. Some researchers point to the positive effect of small dams on the increase in the abundance and biodiversity of macrophytes [58]. Research conducted on ponds in Scotland showed an increase in the species richness and biodiversity of aquatic macrophytes [59]. Other studies indicate that habitat fragmentation has a negative impact on the occurrence of some plant species [60].

The performed studies showed different hydromorphological conditions of small rivers. The studied sites were classified into different classes of hydromorphological status. Based on the HIR value, seven sites were classified as having good ecological status (2nd class), eight sites as moderate (3rd class), and five sites as having low ecological status (4th class). The average HIR value for habitat types A and B was 0.585 and 0.535, respectively (Figure 1, Table 2). There were statistically significant differences between the A and B habitats (Table 4). The A-type sites were characterized by particularly numerous attributes, indicating the natural character of the watercourse (HQA). The invasion of beavers affected the heterogeneity of the current and of the bottom material, as well as altering the diversity of the elements accompanying the tree stand. The results of numerous studies confirm the impact of coastal zone development and water trophy on the hydromorphological conditions of the river [61]. The good ecological status of rivers is related to the occurrence of a semi-natural coastal zone [62]. In international literature, models of the multidimensional dependence of the biodiversity of macrophyte communities on the elements of the river bed morphology are built [15,63]. Our work presents the linear regression equation between HIR and MIR for 20 sites (Figure 5). Due to the small number of stations and the limited range of parameter variability, this relationship is statistically insignificant. The increase in the MIR value is related to the increase in HIR. Our results confirm previous studies showing the influence of hydromorphological conditions on the macrophyte community [15,64]. In addition, an article by Shah et al. [15] showed that the species richness of macrophytes was associated with an increase in the HQA index, representing a measure of the naturalness of the river bed. In our research, the value of the HQA sub-index varied depending on the type of habitat. In habitats of type A, the HQA index ranged from 29 to 49, while in habitats of type B it ranged from 21 to 33. In habitats of type A, the heterogeneity of the current and bed material as well as the diversity of elements accompanying the stands increased. The activity of beavers caused changes in the type of current from laminar to rapid. Another observed change was the accumulation of sediment at the bottom of the river bed. Moreover, the presence of

fallen trees and wood rubble was found. The lowering of the HIR value at some sites is influenced by the HMS sub-index, which ranges from 0 to 30. Its increased value results from the anthropogenically transformed cross-sections of the slopes, the profiling of the bed bottom and the straightening of the bed. Earlier studies have shown that the ecological status of rivers is mainly influenced by the biodiversity of macrophytes. This element of water quality assessment is supported by the hydromorphological state and chemical parameters [16,63,65].

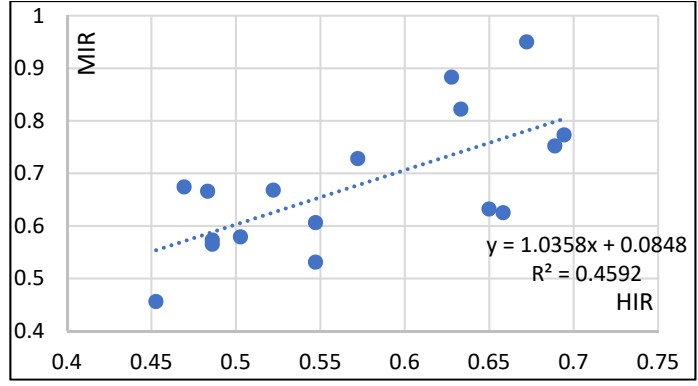
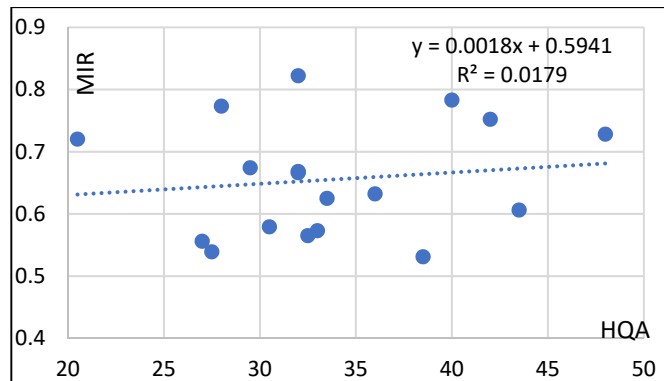

**Figure 5.** Correlation between HIR and MIR as HQA and MIR.

For all parameters, a positive correlation coefficient value was obtained; only for the water level was the coefficient close to zero. In the case of the following variables: HIR, MIR, Shannon–Wiener, and Simpson index, it was found that the correlation for A and B sites was significantly different from zero, with the coefficient exceeding 0.66 (significance level 0.05) (Table 3). On the basis of the t-test, it was found that the average values of HIR and water level differed significantly for sites of type A and B (significance level 0.05). In the case of the remaining parameters, no significant differences were found between the sites located in the areas of the European beaver's occurrence and areas without these aquatic animals' activity (Table 4).

## 4. Conclusions

The conducted research shows that as a result of their activity, wild animals influence a number of changes in the environment. The beaver invasion contributed to significant positive changes in the river's hydromorphological conditions and water levels. Beaver dams also contributed to the increase in the abundance and biodiversity of macrophytes. The activity of beavers contributed to an increase in the share of moisture-loving species, and the systematic replacement of species characteristic of forests with marsh and rush species. On the basis of MIR, 55% of the analyzed river sites were classified as having a moderate ecological status, and the remaining sites met at least the standards of good ecological status. The ecological status was negatively affected by the presence of species characteristic of hypertrophic habitats. Much lower results were achieved for HIR. In habitats of type A, beaver activity caused changes in the type of current from laminar to rapid. Another observed change was the accumulation of sediment at the bottom of the river bed. Moreover, the presence of fallen trees and wood rubble was identified. The reduced HIR value results from the anthropogenically transformed cross-sections of the slopes, the profiling of the bed bottom, and the straightening of the bed. As a result, only 35% of the sites examined were classified as having good ecological status, while the remaining sections did not meet this standard.

**Author Contributions:** Conceptualization, K.P. and A.G.; theoretical discussion, K.P., U.B.-M. and A.G.; methodology, K.P. and M.T.-K.; data analysis, K.P., A.G. and U.B.-M.; writing—original draft preparation, K.P.; writing—review and editing, M.T.-K. and A.G. All authors have read and agreed to the published version of the manuscript.

**Funding:** This paper was written on the basis of the research projects funded by: Polish Ministry of Science and Higher Education—project entitled: Water, wastewater and energy management (contracts No. TKD/DS1, TKD/S/1); The paper was written within the framework of a PhD thesis prepared by Katarzyna Połeć, M.Sc., from the project no SD/27/ISGiE/2021 financed by the University of Life Science in Lublin (Poland) entitled "The influence of the European beaver (Castor fiber) activity on the shaping of water resources".

**Data Availability Statement:** All data generated or analyzed during this study are included in this published article.

**Conflicts of Interest:** The authors declare no conflict of interest.

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
