# Peer review of "Changes in the Ecological Status of Rivers Caused by the Functioning of Natural Barriers"

_water, doi:10.3390/w14091522_

Round 1
Reviewer 1 Report
Review on: Changes in the ecological status of rivers caused by the functioning of natural barriers
The article is dealing with the effect of beavers on the catchment and ecological quality of the river. This is an important aspect from a conservation point of view also. It highlights the positive effect of beavers on river quality by stabilizing water levels.
The methods used in the study are adequate to answer the scientific question..s
I found only small minor corrections and remarks.
L 63 Are there any negative effects of the beaver in Poland?
L 122 "To calculate them, the Braun-Blanquet scale was transformed into percentages." - describe how there are several ways to transform or rescale Braun-Blanquet.
L 160-166 No need to describe box-plot. Refer to the values in the figure captions.
L 172, Pearson correlation is a parametric correlation, did you check normal distribution?
L 286 Describe box-plot here. Also if significant differences occur highlight them with 'a-b' labels.
l 289 Tow decimals would be sufficient.
l 296 the figure caption should describe that what are the numbers and how to resolve the abbreviation names.
L 297-300 Maybe an RDA would be better both to present the different groups the main taxa and the effect of the HIR and WL on the community.
Reviewer 2 Report
The paper needs major modifications before it is processed:
(1) The main findings of this research can be merged into abstract.
(2) What are motivation, innovation, and research organization?
(3) There are five ecological indices which were investigated by for streams throuogth Poland. Author can refer these researches in the literature review.
Non-Linear Visualization and Importance Ratio Analysis of Multivariate Polynomial Regression Ecological Models based on River Hydromorphology and Water Quality
(4) Authors can search possibility of computing N, D, RMNI,IBMR, MIR, HMS, and HQA with their own field data.
(5) Results and Discussions are poorly-written.
(6) Conclusion section should be expanded a lot.
(7) Newly-advanced analysis of water quality parameters for natural stream are found in Natural Resources Research 30 (5), 3761-3775, 2021; Artificial Intelligence Review 54 (6), 4619-4651, 2021.
Round 2
Reviewer 2 Report
Accept as is